# MglA functions as a three-state GTPase to control movement reversals of *Myxococcus xanthus*

Christian Galicia [1], Sébastien Lhospice[2,6], Paloma Fernández Varela [3,6], Stefano Trapani [4], Wenhua Zhang[1,5], Jorge Navaza[1], Julien Herrou[2], Tâm Mignot[2] & Jacqueline Cherfils [1,3]*

In *Myxococcus xanthus*, directed movement is controlled by pole-to-pole oscillations of the small GTPase MglA and its GAP MglB. Direction reversals require that MglA is inactivated by MglB, yet paradoxically MglA and MglB are located at opposite poles at reversal initiation. Here we report the complete MglA/MglB structural cycle combined to GAP kinetics and in vivo motility assays, which uncovers that MglA is a three-state GTPase and suggests a molecular mechanism for concerted MglA/MglB relocalizations. We show that MglA has an atypical GTP-bound state (MglA-GTP*) that is refractory to MglB and is re-sensitized by a feedback mechanism operated by MglA-GDP. By identifying and mutating the pole-binding region of MglB, we then provide evidence that the MglA-GTP* state exists in vivo. These data support a model in which MglA-GDP acts as a soluble messenger to convert polar MglA-GTP* into a diffusible MglA-GTP species that re-localizes to the opposite pole during reversals.

[1] Laboratoire de Biologie et Pharmacologie Appliquée, CNRS and Ecole Normale Supérieure Paris-Saclay, Cachan, France. [2] Laboratoire de Chimie Bactérienne, CNRS-Université Aix-Marseille, Marseille, France. [3] Laboratoire d'Enzymologie et Biochimie Structurale, CNRS, Gif-sur-Yvette, France. [4] Centre de Biochimie Structurale (CBS), INSERM-CNRS-Université de Montpellier, Montpellier, France. [5] Present address: School of Life Sciences, Lanzhou University, Lanzhou, China. [6] These authors contributed equally: Sébastien Lhospice, Paloma Fernández Varela. *email: jacqueline.cherfils@ens-paris-saclay.fr

Small GTPases function as molecular switches in all living organisms by alternating between two states, one that is inactive and bound to GDP, and one that is bound to GTP and recruits effectors (reviewed in ref. [1]). In general, GDP/GTP alternation is finely controlled by guanine nucleotide exchange factors (GEFs), which stimulate GDP/GTP exchange, and by GTPase-activating proteins or GAPs, which stimulate GTP hydrolysis (reviewed in ref. [2]). Polarity of migration controlled by small GTPases is prominent in eukaryotes (reviewed in ref. [3]), and has also been discovered in *Myxococcus xanthus*, a gram-negative deltaproteobacteria that moves by gliding on solid surfaces under the control of the small GTPase MglA and its GAP MglB (reviewed in ref. [4]). Directed movement is essential for *M. xanthus* multicellular differentiation and behavior (reviewed in ref. [5]), and it uses two distinct motility machineries. The Social-motility (S-motility) complex, formed by type-IV pili, allows the collective motion of large groups of cells (reviewed in ref. [6]). In contrast, single motile cells at the periphery of the colony are propelled by the Adventurous-motility (A-motility) system comprised of the Agl-Glt motility complex (reviewed in ref. [7]). Remarkably, both systems are assembled at the bacterial pole under the control of a single small GTPase, MglA[8,9]. In the case of A-motility, MglA-GTP assembles the Agl-Glt complex and recruits its components directly to the MreB actin cytoskeleton, which allows the system to travel towards the lagging pole where it is disassembled by inactivation of MglA by MglB[10,11]. How MglA controls the S-motility system remains to be established. A hallmark of all motile *M. xanthus* cells is the accumulation of MglA-GTP at the leading pole where it activates motility, while MglA-GDP is found diffusely in the cytosol[8,9,12]. This localization pattern arises due to the GAP activity of MglB, which is located at the lagging pole where it inactivates MglA-GTP and depletes it from this pole[8,9]. The ability of MglA to hydrolyze GTP is essential for motility regulation. Notably, locking MglA in the GTP-bound form by mutation of residues that are needed for GTP hydrolysis (e.g. Arg 53 or Gln 82) or by deletion of MglB provokes a characteristic pendulum A-motility regime, in which cells move by exactly one cell length between reversals[8,10]. This remarkable behavior is due to impaired Agl-Glt disassembly at the lagging pole, enabling the complex to continue movement in the opposite direction. The RomR-RomX complex (RomRX hereafter) is a major determinant of MglA-GTP polar targeting, which has been proposed to result from a dual GEF/effector activity[13].

Inversions of the direction of movement (reversals) are characterized by the inter-dependent pole-to-pole oscillations of MglA, MglB and RomRX, which are controlled by signals transduced via the Frz receptor-kinase complex (reviewed in ref. [14]). During movement, MglA and MglB remain segregated at opposite poles, while RomRX steadily relocates from the leading pole to the lagging pole, in a manner that the duration of its complete relocation defines the minimal period between reversals[13,15]. Once RomRX has fully relocated to the lagging pole, signals depending on phosphorylated FrzX, a substrate of the FrzE kinase that functions as a gate at the lagging pole, provoke the rapid detachment of MglA from the leading pole and its relocalization to the opposite pole where it colocalizes transiently with MglB. This colocalization correlates with a pause in movement, the duration of which coincides with the time needed for relocalization of MglB to the opposite pole, eventually allowing movement to resume in the opposite direction. In contrast, the slow relocalization of RomRX is not regulated by signals and starts as soon as the cell initiates a reversion event. This regulatory design functions like a so-called gated relaxation oscillator, allowing the motility switch to adjust to signal levels[15]. Intriguingly, although MglA, MglB and RomRX bind directly to

each other and RomRX facilitates the polar localization of MglA-GTP[13,16,17], MglA-GTP remains at the leading pole when RomRX and MglB occupy the lagging pole. Currently, the mechanism whereby MglA perceives information from the lagging pole at the onset of a reversal leading to its relocalization is not understood.

MglA and MglB are conserved in a large number of bacteria, where they are often encoded by a single operon[18]. Structural analysis of MglA-GDP and MglA-GTP-MglB from the extremophile *Thermus thermophilus* highlighted intriguing features at the switch 1 and 2 regions, which are canonical regions of small GTPases that bind nucleotides[12]. Notably, the switch 1 region undergoes a twisted 3-residue register shift between the GDP- and GAP-bound states, which moves catalytic residues in and out of register to stimulate GTP hydrolysis. Likewise, the switch 2 region departs from the flexible conformation generally observed in GDP-bound GTPases (reviewed in ref. [19]). In MglA-GDP, it adopts a well-ordered conformation that occludes the binding site of the γ-phosphate of GTP, and is displaced by 5 Å to bind GTP in the MglA-GTP-MglB complex[12]. The structure of unbound MglA-GTP is currently unknown, leaving open the question of whether the substantial remodeling at switch 1 and switch 2 is solely due to GTP or is promoted by MglB. MglB also differs from canonical GAPs, which insert a residue, often an arginine, near GTP to complete the catalytic sphere and stabilize the transition state of the hydrolysis reaction (reviewed in ref. [2]). Instead, MglB forms no direct contact with GTP but indirectly positions a switch 1-borne arginine in a catalytic position[12], in a manner reminiscent of the RGS GAPs of eukaryotic heterotrimeric G proteins[20]. While important features of the MglA/MglB cycle were established by these structures, how they support polar localization of MglA-GTP during movement and its rapid detachment at the onset of a reversal is currently not understood.

In this study, we uncover properties of MglA and MglB that provide important mechanistic insight into their oscillatory behavior, by combining the determination of the full structural cycle of *M. xanthus* MglA and MglB, GAP kinetics reconstituted from purified proteins and in vivo motility assays. Our study reveals that MglA has at least three structural and biochemical states, including a mixed inactive/active conformation never observed in a small GTPase, and two GTP-bound states, one of which is refractory to GTP hydrolysis by MglB (MglA-GTP* hereafter). Remarkably, MglA-GTP* can be reverted to the MglB-sensitive MglA-GTP form by MglA-GDP, uncovering a positive feedback mechanism. The existence of an MglB-refractory MglA-GTP* species in vivo is supported by the inability of a diffusible MglB mutant, which is able to reach all MglA-GTP species in the bacteria, to displace the polar MglA-GTP species. Together, our findings suggest that the MglA-GTP cluster located at the leading pole is composed of two MglA-GTP populations, including a major MglA-GTP* species that is resistant to MglB-stimulated GTP hydrolysis and is predicted to have the mixed active/inactive conformation, and a minor diffusible MglA-GTP species that can interact with the motility machineries and be inactivated by MglB at the lagging pole. When cells reverse, a rapid increase of the MglA-GDP pool could therefore convert the entire polar pool of MglA-GTP* into a diffusible MglB-sensitive MglA-GTP form, thus explaining how the decision to reverse is conveyed from the lagging pole to the leading pole.

## Results

**MglA has a three-state GDP/GTP structural switch.** To gain insight into the structural landscape of *M. xanthus* MglA, we determined the crystal structures of MglA in different nucleotide states, which uncovered that MglA can adopt three

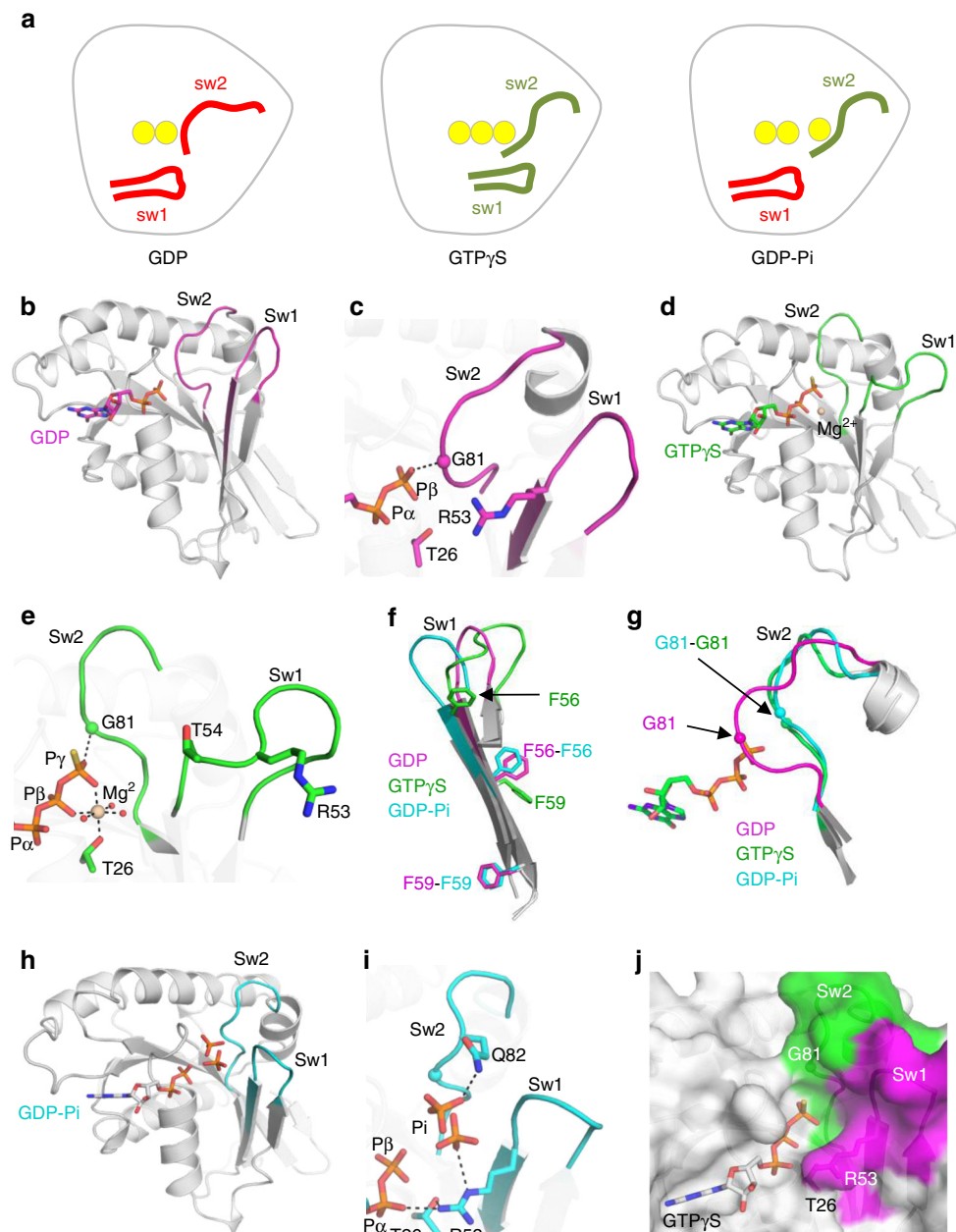

**Fig. 1** MglA has a three-state GDP/GTP switch. **a** Schematic representation of the GDP-bound, GTP-bound and mixed MglA conformations. The inactive conformations of the switch regions are in red, the active conformations are in green. The phosphates of GDP, GTP and Pi are depicted as yellow spheres. **b** Structure of MglA-GDP. The switch 1 is retracted and switch 2 has an autoinhibitory conformation. **c** Close-up view of nucleotide-binding site of MglA-GDP. The arginine finger is remote from the nucleotide and Pro80 and Gly81 from switch 2 occlude the $Mg^{2+}$ and γ-phosphate-binding sites. **d** Structure of MglA-GTPγS-$Mg^{2+}$. Switch 1 and switch 2 have active conformations. **e** Close-up view of the nucleotide-binding site in MglA-GTPγS. The arginine finger is remote from the nucleotide and $Mg^{2+}$ has an incomplete coordination sphere. **f** Overlay of switch 1 from MglA-GDP, MglA-GTPγS and the mixed MglA form (see Fig. 1g, h for this form). The twisted three-residue shift in the GTPγS-bound form is highlighted by the positions of F56 and F59. Note the flexibility of switch 1 loop. **g** Overlay of switch 2 in MglA-GDP, MglA-GTPγS and the mixed MglA form, showing the large difference near Gly 81. Only GTPγS is shown for clarity. **h** The mixed MglA form features a retracted switch 1 and an active switch 2 conformations. The structure contains GDP and a phosphate or sulfate. **i** Close-up view of the nucleotide-binding site of the mixed MglA form. The partially occupied phosphate/sulfate positions are shown. **j** The mixed MglA form can accommodate GTP. Overlay of GTPγS taken from the MglA-GTPγS structure onto the mixed MglA structure. The surface of the nucleotide-binding site is shown as a surface, with the retracted switch 1 in pink and the active switch 2 in green. Color-coding for all panels except (a) and (j) is: GDP-bound: pink; GTPγS-bound: green; mixed form: blue. Hydrogen bonds are shown as dotted lines. $Mg^{2+}$ is shown in beige. Sw1: switch 1. Sw2: switch 2.

conformations: inactive MglA-GDP, active MglA-GTPγS, whose structure had not been determined before, and an unusual conformation combining inactive and active features (Fig. 1a, Table 1 and Supplementary Table 1). In MglA-GDP, the inactive switch 1 is retracted, which positions the catalytic arginine finger (Arg 53)

remote from the nucleotide, and switch 2 has an autoinhibitory conformation that occludes the binding sites of GTP and $Mg^{2+}$ (Fig. 1b, c), as seen in *T. thermophilus* MglA-GDP[12]. Binding of GTPγS induces a twisted three-residue register shift of switch 1 and remodels switch 2 to create space for the γ-phosphate, as seen

**Table 1 Crystal structures determined in this study.**

| Protein | Space group | Resolution | PDB code |
|---|---|---|---|
| MglA-GDP | $P\,2_1\,2_1\,2_1$ | 1.98 | 6HJO |
| MglA-GTPγS | $I\,2\,3$ | 1.28 | 6H17 |
| MglA mixed form | $P\,2_1\,2_1\,1$ | 2.3 | 6H35 |
| MglA-GTPγS-MglB | $P\,6_4$ | 2.8 | 6H5B |
| MglB | $P\,2_1\,2_1\,2_1$ | 2.39 | 6HJM |

in MglA in the *T. thermophilus* MglA-MglB complex[12] (Fig. 1d, e). The shift of switch 1 results in the elongation of the arginine finger loop (residues 44−58) (Fig. 1f), but Arg53 remains outside the nucleotide-binding site. Displacement of switch 2 also allows $Mg^{2+}$ to bind with an incomplete coordination that lacks the conserved $Mg^{2+}$-binding threonine (Thr54) located in the arginine finger loop (Fig. 1e). Remarkably, we trapped MglA in a mixed state in which switch 1 is in the inactive conformation and switch 2 is in the active conformation, an inactive/active combination never observed in a small GTPase before (Fig. 1f–i and Supplementary Fig. 1A-C). The nucleotide-binding site contains GDP and a ligand tentatively ascribed to inorganic phosphate (Pi) with two partially occupied positions, possibly arising from spontaneous hydrolysis of GTPγS over time. There is no $Mg^{2+}$ in this structure, as its binding site is occluded by the retracted switch 1. Structures of small GTPases with bound GDP-Pi have shown that this state retains the conformation of the GTP-bound precursors[21,22]. Accordingly, overlay with MglA-GTPγS shows that GTP can readily be accommodated in the mixed MglA conformation (Fig. 1j), suggesting that MglA can adopt two distinct GTP-bound states. Together, our crystallographic analysis reveals that MglA has three conformational states, one of which combines a retracted switch 1 and an active switch 2 predicted to derive from an atypical GTP-bound state.

**Crystal and solution structural analysis of MglB**. To complete the MglA GDP/GTP structural cycle, we determined the crystal structures of its GAP MglB in unbound form and in complex with MglA-GTPγS (Table 1 and Supplementary Table 1). The MglB crystal contains 20 molecules in the asymmetric unit, for which we applied an original strategy based on a low-resolution envelope derived from another weakly diffracting crystal form, making it one of the molecular replacement structure determination with the largest number of independent molecules to date (see Methods, Supplementary Fig. 2a, b). The 20 monomers are arranged as 5 tetramers (Supplementary Fig. 2c, d), which resemble tetramers seen in *T. thermophilus* MglB crystal structures[12]. This tetrameric arrangement masks the MglA-binding site and would thus be autoinhibitory. We therefore analyzed the quaternary structure of MglB in solution using SEC-MALS, which shows that the major MglB species in solution is in fact a dimer, in which the active site should thus be available (Supplementary Fig. 2e). As previously observed for *T. thermophilus* MglB, *M. xanthus* MglB subunits have a roadblock fold (Figure S2d), which is frequently encountered in regulators of small GTPases (reviewed in ref. [23]). The C-terminus (residues 133−165) is not visible in any of the 20 independent MglB monomers of the asymmetric unit, indicating that it is highly flexible. To gain further insight into the conformation of this segment, we characterized the structure of MglB in solution using SEC-SAXS (Fig. 2a and Supplementary Fig. 2f–h). The dimensionless Kratky plot is representative of a globular protein with a significant proportion of flexible regions (Supplementary Fig. 2h). The maximal dimension ($D_{max}$) of 96 Å is significantly larger than the $D_{max}$ of the MglB core that is visible in the crystal (73 Å), with the

SAXS envelope showing additional volumes extending the dimeric core on each side (Fig. 2b). Combining the SAXS and crystallographic data, the two C-terminii missing from the crystal structures could be modeled as intrinsically disordered peptides, yielding an excellent fit with the experimental data ($\chi^2 = 1.22$) (Fig. 2c and Supplementary Fig. 2i). Thus, the C-terminus is a flexible segment that enlarges the volume occupied by MglB. Its sequence is conserved across species (Supplementary Fig. 1j), which suggests that it plays a role in MglB functions, possibly exploiting its flexibility.

Next, we determined the crystal structure of the *M. xanthus* MglA-GTPγS/MglB complex. The complex contains a symmetrical MglB dimer and one MglA-GTPγS molecule (Fig. 2d), as previously observed for *T. thermophilus* MglA-MglB complexes[12]. As in unbound MglB, the C-terminii of MglB are disordered in the MglA-MglB complex. Both MglB monomers interact through their helix 2 with switch 1 and switch 2 of MglA, which display active conformations (Fig. 2e, f), with one monomer establishing additional contacts outside the switch regions. The MglB dimer does not insert any element in the nucleotide-binding site and does not interact with GTPγS directly. Overlay of unbound MglA-GTPγS shows that one MglB monomer would clash with the arginine finger loop, a conflict that is resolved by a large lasso movement of switch 1 (11 Å at Arg53) (Fig. 2f). This rearrangement positions the arginine finger near the active site, which it could reach through side chain rotation, and it completes the canonical coordination of $Mg^{2+}$ by the conserved switch 1 threonine (Thr 54) (Fig. 2f). Thus, MglB recognizes the active conformations of the switch regions and contributes to catalysis by reorganizing the arginine finger loop through steric conflict. The organization of the MglA-MglB complex is surprisingly similar to that of the complex of the Rab GTPase Ypt1 with the TRAPP complex, a yeast RabGEF whose central subunits have roadblock folds (Supplementary Fig. 2k)[24]. Therefore, we tested whether MglB might also act as a GEF for MglA, by measuring nucleotide exchange using fluorescence kinetics. As shown in Supplementary Fig. 2l, the slow spontaneous GDP/GTP exchange of MglA was not increased by MglB, indicating that MglB does not function as a GEF under these conditions.

**Feedback regulation of an MglB-resistant MglA-GTP form**. The identification of an atypical mixed MglA structure raises the question of whether such a mixed state could affect MglB-stimulated GTP hydrolysis. To get insight into this issue, we characterized the kinetics of MglB-stimulated GTP hydrolysis stimulated by fluorescence, using an engineered bacterial phosphate-binding protein[25,26]. Potent GAP activity was measured with this assay, which depended on the presence of MglB and GTP (Supplementary Fig. 3a). GTP hydrolysis kinetics measured over a range of MglB concentrations yielded a $k_{cat}/K_M$ of $2.1 \times 10^3\,M^{-1}\,s^{-1}$ (Fig. 3a and Supplementary Fig. 3b). While carrying out these experiments, we made the intriguing observation that MglA-GTP samples became gradually less sensitive to MglB over time when incubated at 25 °C and were resistant to GTP hydrolysis after about 1 h (Fig. 3b). In contrast, MglB remained equally active over time, indicating that the effect is entirely comprised in MglA and occurs spontaneously. This effect was also not due to MglA unfolding over time, as checked by circular dichroism (Supplementary Fig. 3c). We refer to this state as MglA-GTP* hereafter.

Next, we investigated whether resistance of this MglA-GTP* state to MglB could be reverted. Remarkably, MglB-resistant MglA-GTP* samples could be fully resensitized to MglB-stimulated inactivation by addition of MglB-sensitive MglA-GTP (Fig. 3c). No resensitization was observed by addition of

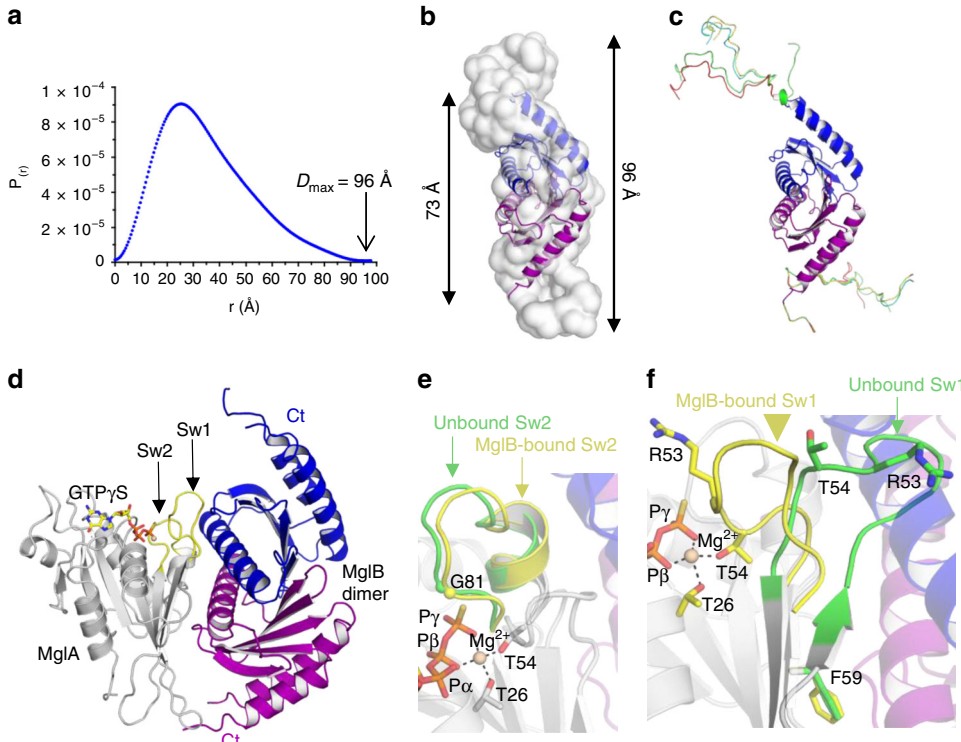

**Fig. 2** Crystallographic and solution structural analysis of MglB. **a** SEC-SAXS distance distribution function $P(r)$ of unbound MglB in solution. The $D_{max}$ value is indicated. Complementary information on SAXS data collection and analysis is given in Supplementary Fig. 2f–i. **b** Comparison of the crystallographic MglB dimer (in blue and purple) with the SAXS envelope calculated with GASBOR. The C-terminii are not visible in the crystal structure, but appear as extra volumes in the SAXS envelope (arrows). The $D_{max}$ values calculated from the crystal structure and the SAXS data are indicated. **c** A composite model of the MglB dimer, in which the C-terminii is represented by ensemble structures calculated with MultiFoXS. **d** Crystallographic structure of the MglA-GTPγS/MglB heterotrimer. MglB interacts with switch 1 and with nonswitch regions of MglA, forming an interface that is asymmetrical on the MglA side and symmetrical on the MglB dimer side. The position of the C-terminii of MglB is not visible in the crystal. **e** Close-up view of the overlay of switch 2 in unbound (green) and MglB-bound (yellow) MglA-GTPγS, showing that the conformation of switch 2 is not remodeled by MglB. **f** Close-up view of the overlay of switch 1 in unbound (green) and MglB-bound (yellow) MglA-GTPγS, showing that MglB induces a lasso movement of the arginine finger loop that resolves steric conflicts and brings Arg53 and Thr54 into the nucleotide-binding site.

GTP alone or of MglA-GTPγS, which cannot be converted to MglA-GDP, while addition of MglA-GDP fully resensitized MglA-GTP* (Fig. 3d). These observations indicate that the resensitizing species is MglA-GDP, which in the experiment shown in Fig. 3c is produced by inactivation of MglA-GTP by MglB. The observation that the mixed MglA conformation observed in the crystal contains GDP and Pi raises the possibility that the MglB-refractory MglA species is a post-hydrolysis intermediate resulting from spontaneous hydrolysis of GTP into GDP-Pi. However, analysis of the nucleotide content of MglB-resistant MglA samples showed that the major bound nucleotide was GTP (Supplementary Fig. 3d, e). Thus, MglA-GTP* contains GTP, and its resistance to MglB-stimulated inactivation is not due to the slow release of Pi being a critical rate-limiting post-hydrolysis step.

We conclude from these experiments that MglA-GTP exists in two biochemical states: one that is sensitive to MglB and one that is refractory to MglB and can be resensitized by MglA-GDP produced by MglB through a positive feedback loop.

**An MglB-resistant MglA-GTP population exists in *M. xanthus*.** Our structural and in vitro results suggest that MglA can adopt an MglB-refractory MglA-GTP* state, which raises the question of the existence of this species in vivo. In *M. xanthus*, MglA-GTP accumulates at the leading pole while MglA-GDP is found in the cytosol[8,9,12]. Accordingly, we predict that an MglB-refractory

MglA-GTP* species should be insensitive to the action of a cytosolic MglB variant able to reach all MglA-GTP in the cell, and thus should remain bound to the pole in the presence of such variant. To generate such an MglB variant, we analyzed our crystal structures of MglB and of the MglA/MglB complex for mutations that could render MglB cytosolic without impairing its GAP activity. The MglB dimer features an extended, positively charged, convex tract located opposite to the MglA-binding site (Fig. 4a) which is conserved across bacterial species (Supplementary Fig. 2j), hence appearing well-suited to support intermolecular interactions that determine MglB segregation at the lagging pole. We mutated K14, K120 and R115 in this tract into alanines (MglB³ᴹ) (Fig. 4a), resulting in the removal of six positively charged residues from the surface of the MglB dimer. MglB³ᴹ was as efficient as MglBᵂᵀ at inactivating MglA-GTP in vitro (Supplementary Fig. 4a), indicating that the tract is not involved in GTP hydrolysis and confirming that, similar to MglBᵂᵀ, MglB³ᴹ does not require additional components to inactivate MglA. In vivo, MglB³ᴹ fused to the Neon-Green protein was stably expressed (Supplementary Fig. 4b), and displayed a conspicuous diffuse pattern that contrasted with the mostly unipolar distribution of MglBᵂᵀ (Fig. 4b). These experiments thus identify the region in MglB that supports its recruitment at the lagging pole. Since this region does not overlap with the MglA-binding site, the localization and the GAP activity of the MglB³ᴹ mutant are effectively uncoupled.

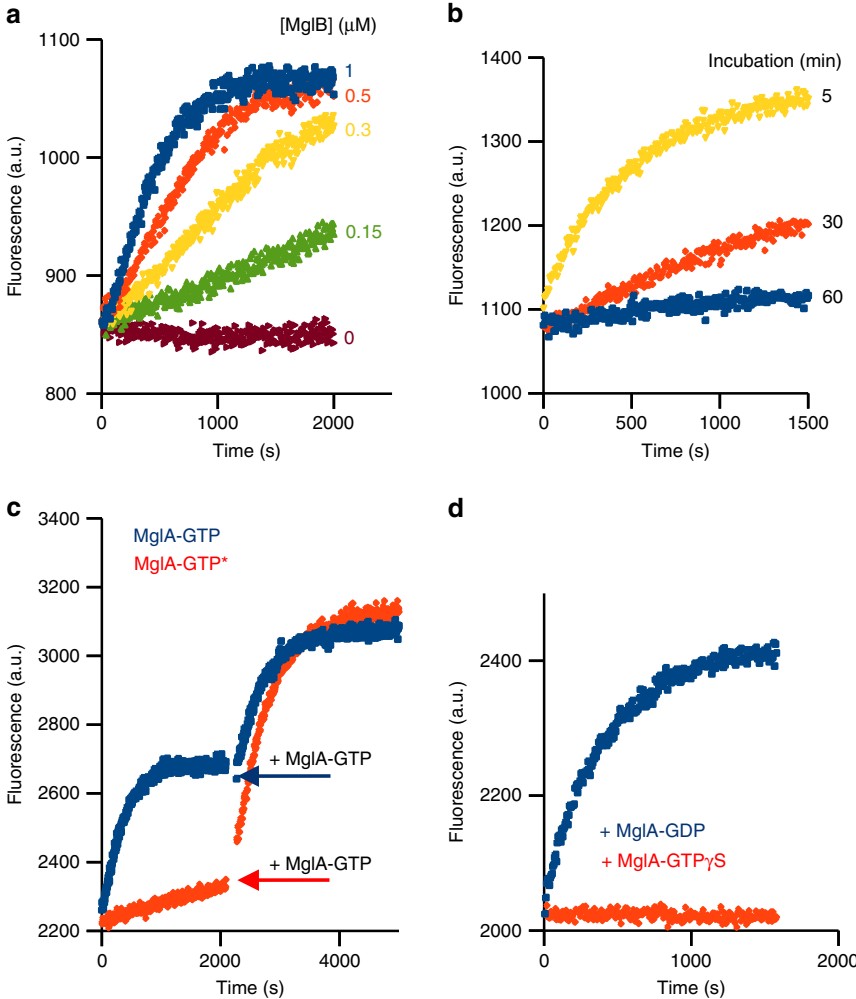

**Fig. 3** Inactivation of MglA by MglB is regulated by a positive feedback loop. **a** Kinetics of GTP hydrolysis stimulated by MglB. Kinetics were monitored by fluorescence using a reagentless phosphate-binding assay at a fixed MglA concentration (1 μM) and increasing concentrations of MglB as indicated. **b** Time-dependent desensitization of MglA to MglB-stimulated GTP hydrolysis. MglA-GTP was incubated at 25 °C for increasing periods of time as indicated, before GTP hydrolysis was initiated by addition of MglB. **c** Desensitization of MglA from MglB is reversed by addition of MglB-sensitive MglA-GTP. MglB was added to MglB-insensitive (in orange) or MglB-sensitive (in blue) MglA-GTP and GTP hydrolysis was monitored for the duration needed for the MglB-sensitive sample to reach the plateau. Then, the same amount of MglB-sensitive MglA-GTP was added to both samples (arrow). Note that both kinetics curves reach the same plateau, indicating that all MglA-GTP has been converted to MglA-GDP in all experiments. **d** MglA-GDP, but not MglA-GTPγS, re-sensitizes MglA-GTP to MglB. MglA-GTP was incubated at 25 °C for 30 min, then MglB and 1 μM of either MglA-GDP or MglA-GTPγS was added.

The above experiments show that MglB[3M] has the required functional features to investigate whether all MglA-GTP, which has a mostly polar localization, can be converted by MglB into MglA-GDP, which is characterized by a cytosolic distribution[9,10,12]. Remarkably, MglA-YFP was still able to localize at the poles in MglB[3M]-expressing cells (Fig. 4c and Supplementary Fig. 4c), suggesting that a significant fraction of MglA-GTP is resistant to MglB-stimulated GTP hydrolysis even under conditions where the two proteins are no longer segregated from each other. This result could be explained (i) if polar MglA-GTP (or a fraction of it) is insensitive to MglB inactivation (i.e. MglA-GTP*) or (ii) if MglB[3M] cannot reach MglA at the pole. To discriminate between these possibilities, we took advantage of a mutation in switch 2 of MglA (Q82L), which renders it unable to hydrolyze GTP. MglB binds stably to MglA[Q82L]-GTP, thus competing with the motility machinery for binding MglA and thereby blocking motility[10]. In this background, deletion of *mglB* restores pendulum movements that are characteristic of a constitutive MglA activity[10] (Fig. 4d). Thus, the frequency of

pendulum movement reports on the interaction of MglA[Q82L] with MglB. We therefore compared the frequency of pendulum movements induced by MglA[Q82L] in MglB[3M]-expressing cells to that in the Δ*mglB* mutant. Pendulum movements were conspicuously reduced by the diffusible MglB[3M] mutant compared to the Δ*mglB* mutant (Fig. 4d), which rules out that MglB[3M] behaves as a deletion mutant and confirms that it is able to bind MglA in vivo. Thus, the polar MglA-GTP cluster is comprised of an MglB-resistant MglA-GTP* species.

Since *Myxococcus* cells expressing MglB[3M] appear to accumulate MglA-GTP*, we could assess the functional relevance of the balance between MglA-GTP and MglA-GTP*. Strikingly, single cells expressing MglB[3M] displayed highly reduced motility, and this defect was more severe than that resulting from *mglB* deletion (Fig. 4e and Supplementary Fig. 4d). This observation suggests that the MglB-resistant MglA-GTP* subpopulation cannot engage productively with the motility complex. Moreover, although fluorescence fluctuations were observed at the cell pole, polar switching of MglA-YFP as occurs during reversals of

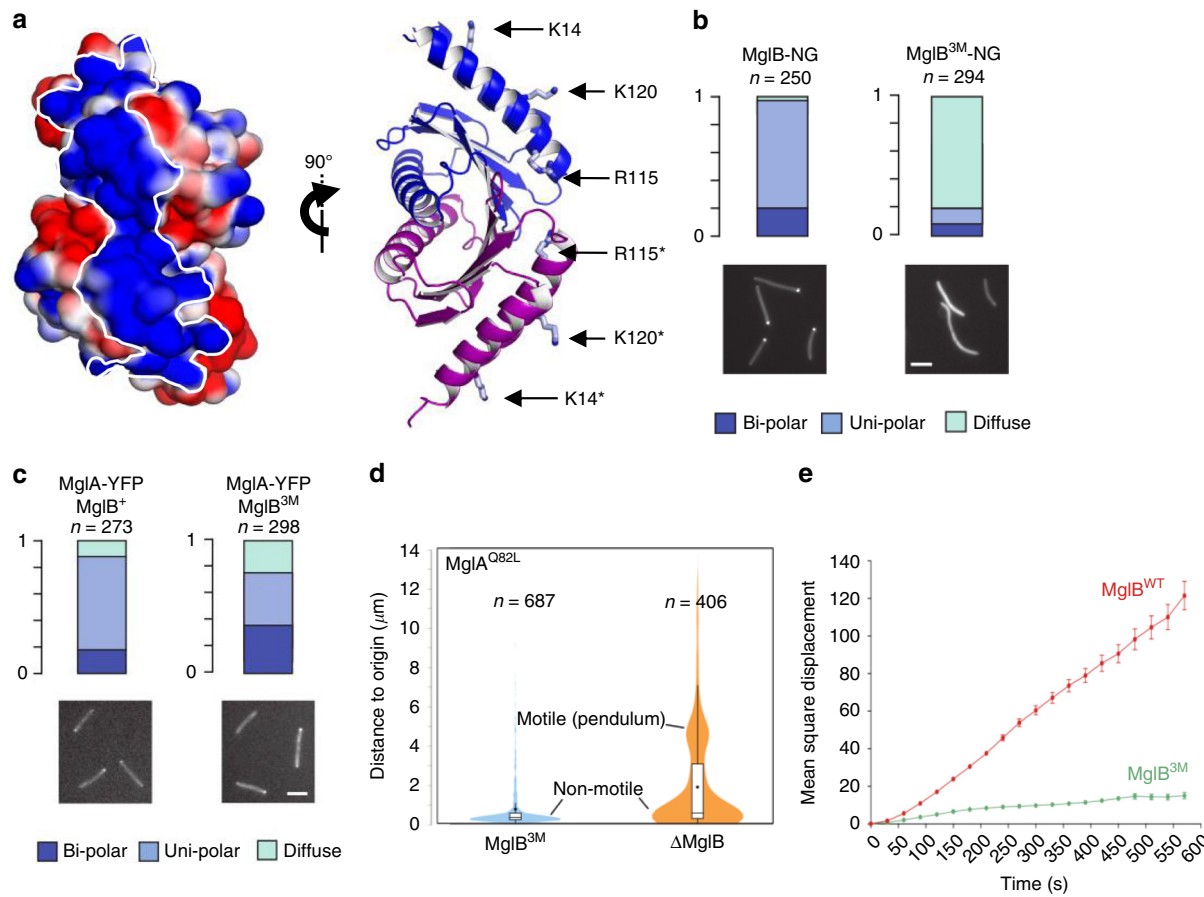

**Fig. 4** The MglB positively charged tract determines the polar localization of MglB. **a** MglB features a positively charged tract opposite to the MglA-binding site. The left panel shows electrostatic potential calculated with APBS[50]. The tract is delineated by a white line. The right panel shows the residues in this tract mutated in MglA[3M]. The two views are rotated by 90°. **b** MglB[3M]-NG has a diffuse localization. Localizations of MglB-NG and MglB[3M]-NG are shown. Scale bar = 2 μm. **c** MglA-YFP retains polar localization in MglB[3M]-expressing cells. Localization of MglA-YFP in cells expressing MglB or MglB[3M] is shown. Scale bar = 2 μm. **d** MglB[3M] blocks the action of a GTP-locked MglA mutant. The motility of MglA[Q82L]-expressing cells was measured in *mglB* deletion or in *mglB[3M]*-expressing strains. For each cell, motility is measured as the distance traveled from the origin for a time period of 45 min. Note that MglB[3M]-expressing cells are mostly nonmotile while MglB[WT]-expressing cells show two distinct populations, one nonmotile and one motile with reduced traveled distance to origin due to previously described pendulum movements. **e** Expression of MglB[3M] blocks motility. Motility is measured as the mean square displacement (MSD) of single cells on agar. The average MSD of *n* trajectories and associated standard error of the mean are shown for the wild-type strain (*n* = 2658) and the MglB[3M] strain (*n* = 3616). MglB[WT]: *DZ2 ΔmglB pSWU19-mglB[WT]*, MglB[3M]: *DZ2 ΔmglB pSWU19-mglB[3M]*. Source data for panel (**e**) are provided as a Source Data File.

WT cells was never observed (Supplementary Fig. 4c), indicating that both MglA-GTP species together with polar MglB are required for the polarity switch.

Together, these observations provide strong evidence that an MglB-resistant MglA-GTP* subpopulation exists in vivo, and that this MglA-GTP* species serves a polarity function but is unable to support motility.

## Discussion
In this study we investigated how the structural and biochemical features of the GDP/GTP switch of the small GTPase MglA contribute to molecular oscillations that control reversals in the gliding bacterium *M. xanthus*. We discovered that MglA has more than two structural, biochemical and functional states, which were consistently observed (1) by X-ray crystallography, uncovering a mixed usage of active and inactive switch region conformations never seen in a small GTPase before, (2) by kinetics analysis, revealing that MglA-GTP converts from an MglB-sensitive to an MglB-resistant form (coined MglA-GTP*), in a manner that is reversed by a positive feedback loop operated

by MglA-GDP, and (3) in live bacteria, using a diffusible MglB mutant, which reveals the existence of an MglB-resistant MglA-GTP* population that localizes to the cell poles but cannot support motility.

Several lines of evidence suggest that the MglA-GTP* species identified biochemically and in vivo corresponds to the mixed inactive/active conformation seen in the crystal: (1) considering that all known crystal structures of small GTPases bound to GDP-Pi retain the conformation of their GTP-bound precursor[21,22], and that the mixed structure, although it contains GDP-Pi, can readily accommodate GTP (Fig. 1j), it is likely that the mixed structure derives from a GTP-bound MglA species with the same mixed conformation. (2) The retracted conformation of switch 1 in the mixed form does not allow it to establish the interactions seen in the MglA/MglB complex, which can readily explain its resistance to MglB-stimulated GTP hydrolysis. (3) Similar to eukaryotic small GTPases (reviewed in ref. [27]), dissociation of Pi is not significantly rate-limiting for MglB activity and a GDP-Pi intermediate does not accumulate, making it unlikely that the MglA cycle involves a functional post-hydrolysis GDP-Pi intermediate. Accordingly, the MglB-sensitive

MglA-GTP species is likely represented by the MglA-GTPγS structure, in which both switch regions are in active conformations similar to those seen in the MglA/MglB complex (this study, [12]). From a structural perspective, it is remarkable that the unique twisted retraction of switch 1, previously thought to implement a canonical two-state cycle, is thus in fact the signature of a three-state GDP/GTP switch. Our results show that the MglB$^{3M}$ mutant provides a genetic tool to investigate the in vivo relevance and functions of the three MglA forms. Future studies are now needed to identify mutations that trap the MglA-GTP* form to further investigate its function in cells. It can be anticipated that such mutations will have to target allosteric sites located outside the nucleotide-binding site, as mutations within the switch regions are predicted to affect all forms.

Our study suggests that distinct functional MglA-GTP sub-populations operate in gliding bacteria: one that clusters stably at the leading edge to define polarity during movement, and one that associates with the motility complex and travels backwards to propel the bacterium. Because of their structural differences at the switch 1 region, MglA-GTP and MglA-GTP* have the potential to be discriminated by effectors and regulators, hence to define such subpopulations. Since polar MglA-GTP* is not sufficient for motility, it is predicted to bind to signaling components involved in defining the leading pole. Likewise, MglA-GTP has features that are predicted to allow it to recruit the motility complex, become transported to the lagging pole and be inactivated by MglB. Future studies are needed to determine whether MglA effectors and regulators bind in an exclusive manner to a specific MglA-GTP species and/or if they can bind either productively or unproductively depending on the MglA-GTP species. The regulation of this three-state GDP/GTP switch by a positive feedback loop operated by MglB and MglA-GDP potentially explains how MglA rapidly detaches from the leading pole to complete the reversal process (Fig. 5). In this model, reloading of MglA-GDP with GTP is catalyzed by RomRX. In vitro, the spontaneous conversion of MglA-GTP into MglA-GTP* is relatively slow, but could be facilitated in vivo through equilibrium displacement as MglA-GTP* binds to specific effectors. Eventually, conversion of MglA-GTP* into MglA-GTP is controlled by the MglA-GDP pool generated by inactivation of MglA-GTP by MglB (see also below). Importantly, resensitization of MglA-GTP* by MglA-GDP is rapid in vitro, as expected for an efficient feedback effect in vivo. Thus, MglA-GDP could be the mobilization messenger that converts polar MglA-GTP* into diffusible MglA-GTP, allowing it to reach the lagging pole to be inactivated by MglB during reversals. This positive feedback loop would explain the fast, nonlinear dynamics of MglA relocalization to the lagging cell pole when reversals are provoked by Frz signaling[15]. Further work is needed to prove that the feedback loop operates in vivo, but the observation that an MglA-YFP population remains immobilized at the cell pole when the diffusible MglB$^{3M}$ mutant is expressed strongly suggests that the MglA-GDP/MglA-GTP*/MglA-GTP cycle is required for functional MglA oscillations. Interestingly, the mixed MglA structure and MglA-GDP interact in the crystal through their retracted switch 1 (Supplementary Fig. 3f), opening the possibility that MglA-GDP form heterodimers with MglA-GTP* to facilitate their conversion into MglA-GTP, which remains to be investigated.

Finally, our study provides insight into the determinants of MglB clustering at the lagging pole. First, it identifies a conserved, positively charged surface as the major binding site for polar segregation of MglB. This region is located opposite to the MglA-binding site, indicating that interactions of MglB that determine its localization are not exclusive to its interaction with MglA. Given the marked electrostatic character of the identified epitope, we asked whether MglB might interact directly with the

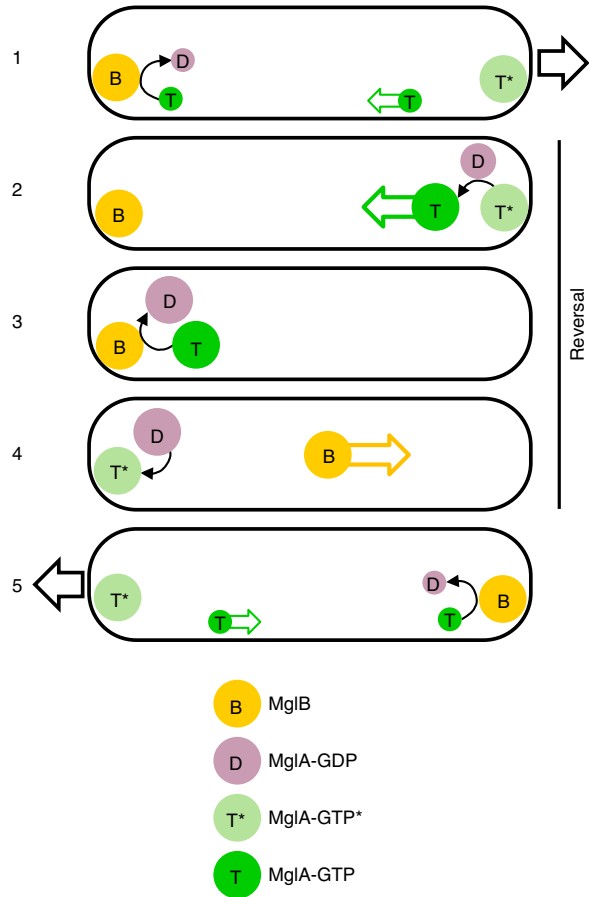

**Fig. 5** Model of the three-state MglA switch. 1 The MglA-GTP cluster located at the front pole of the gliding bacteria is composed of a major species stably associated to the pole which is resistant to MglB (MglA-GTP*, T* in kaki) and a minor diffusible species (MglA-GTP, T, in green) that can reach the lagging pole to be inactivated into MglA-GDP (D in mauve) by MglB (B, in orange). 2 During a reversal event, MglA-GDP is released in the cytosol to reach the leading pole where it converts all polar MglA-GTP* into diffusible MglA-GTP, which detaches rapidly from the pole. 3 MglA-GTP is rapidly inactivated into MglA-GDP by MglB at the lagging pole. 4 MglA-GDP is reloaded with GTP by a GEF activity to regenerate MglB-resistant MglA-GTP*, and MglB relocates to the opposite pole. 5 A new leading pole is established, resuming movement in the opposite direction. The direction of movement at steps 1 and 5 is shown by a black arrow, the reversal event spanning steps 2−3−4 is indicated by a vertical line.

membrane using liposomes containing cardiolipin, a negatively charged lipid which is enriched at bacterial poles (reviewed in ref. [28]), using a cosedimentation assay. However, MglB$^{WT}$ bound only weakly to liposomes, and this weak binding was not measurably affected in MglB$^{3M}$ (Supplementary Fig. 4e). We cannot exclude that the convex shape of the MglB surface could be selective for membranes with negative curvature, which cannot be recapitulated by liposome experiments. Alternatively, or in coordination, MglB localization to the cell pole may require interaction with protein partners, which remain to be identified. Second, we identified the C-terminii of the MglB dimer as intrinsically disordered regions, which enlarge the volume occupied by MglB. The entropic determinants of intrinsically disordered regions are increasingly been shown to support a remarkable variety of functional properties such as substrate recognition[29], enzyme efficiencies[30], membrane curvature sensing[31] and clustering on membranes[32] or through liquid−liquid

phase separations[33]. It is thus plausible that one or several such properties are supported by the disordered C-terminii of MglB and contribute to its clustering and function at the lagging pole, which remain to be investigated. Identification of the positively charged surface and the intrinsically disordered C-terminii of MglB as functionally and/or structurally important regions should now provide a robust framework to investigate their roles in regulated MglB clustering and activity during movement.

The full understanding of the three-state GTPase switch still requires that important remaining issues are elucidated. In addition to those already discussed above, another one is the mechanism that controls the mobilization of the MglA-GDP pool at the onset of a reversal. In one mechanism, MglA-GDP could be generated rapidly by MglB following FrzX signaling. Alternatively, it may be actively retained at the lagging pole and protected from reloading GTP, until a signal releases it into the cytosol. In conclusion, we propose that the existence of three distinct MglA states is a key ingredient to enable MglB, RomRX and MglA to communicate at a distance by means of a diffusible messenger, thereby implementing the spatial component of the relaxation oscillator switch. Given that small GTPases are involved in polarity control, it is likely that such regulation also operates in a number of other biological systems.

## Methods

**Proteins**. All proteins were cloned in a pET28 plasmid. The MglA construct contains a non-cleavable 6xHis-tag at the C-terminus. Two MglB constructs were used. One construct carries a 34-residue tag at the N-terminus including a 6xHis tag and a thrombin cleavage site and was used for the crystal structure of unbound MglB (MglB-1). A second MglB construct carries a 6xHis tag followed by a TEV protease cleavage site at the N-terminus was used for all other structural and biochemical experiments (MglB-2). The MglB$^{K14A/R115A/K120A}$ triple mutant (MglB$^{3M}$) was obtained by inserting each mutation using a specific oligonucleotide starting from the MglB-2 plasmid (Supplementary Table 5).

All proteins were expressed in *E. coli* BL21(DE3) strain grown in LB medium. Overexpression was induced by 1 mM IPTG overnight at 20 °C. Bacteria were pelleted by centrifugation, resuspended in lysis buffer (Tris 20 mM pH 7.5, 300 mM NaCl, 20 mM imidazole, 5 mM DTT) complemented with benzonase and an antiprotease inhibitor cocktail and disrupted in a pressure cell homogenizer at 20,000 psi and centrifuged. Cleared lysates were loaded onto a nickel affinity column (HisTrap FF, GE Healthcare) mounted on an Akta chromatography system. Proteins were eluted by a gradient of imidazole (20−500 mM). Purification of MglA was completed by gel filtration chromatography on a Superdex75 10/300 column (GE Healthcare) equilibrated with a buffer containing Tris 20 mM pH 8, 50 mM NaCl and 10 mM MgCl$_2$. Purification of MglB-1 was completed by gel filtration chromatography on a Superdex75 16/600 column equilibrated with a buffer containing 50 mM Tris pH 8, 100 mM NaCl, 1 mM MgCl$_2$. The 6xHis tag of MglB-2 was cleaved by incubation with the TEV protease overnight at 4 °C (1:20 w/ w ratio). The cleaved tag was removed by a second Ni$^{2+}$ affinity step, then purification was completed by gel filtration chromatography on a Superdex75 10/300 column equilibrated with a buffer containing 20 mM Tris pH 8, 50 mM NaCl and 10 mM MgCl$_2$. MglB$^{3M}$ was purified as MglB-2. All proteins were pure over 90% as judged by SDS-PAGE, concentrated to 2 mg/mL and stored at −20 °C.

Homogeneous loading of MglA with GDP, mant-GDP, GTP or GTPγS was achieved by incubation with a 20× molar excess of either nucleotide for 30 min in a buffer containing 10 mM MgCl$_2$, then excess nucleotide was removed by gel filtration using a Superdex75 column. Incubation was carried out at room temperature, except for GTP which was carried out at 4 °C to prevent spontaneous desensitization to MglB.

**Crystallization and crystallographic analysis**. *MglA-GDP*: Initial crystallization conditions for MglA-GDP were identified with a sparse matrix screen (MPD Suite, Qiagen) by vapor diffusion in sitting drops on a Cartesian crystallization robot. After optimization, diffracting crystals were obtained in 0.2 M sodium tartrate and 40% MPD. Crystals were cryoprotected with 10% glycerol and flash frozen before data collection at beamline PROXIMA 1 (SOLEIL synchrotron, Gif-sur-Yvette, France). Diffraction data were processed and scaled using the XDS package[34].

*MglA-GTPγS*: MglA-GTPγS was supplemented with a 5× molar excess of GTPγS before crystallization. Initial conditions were obtained with the PEGs II Suite screen (Qiagen) by vapor diffusion in sitting drops at 18 °C using a Mosquito robot (TTP Labtech). Diffracting crystals were obtained in 0.2 M ammonium sulfate, 0.1 M sodium acetate pH 4.6, 12% (w/w) PEG 4000 in batch wells containing 1 μL of the sample with 1 μL of crystallization solution covered by paratone and kept at 4 °C. Crystals were cryoprotected by transfer to the reservoir solution supplemented with 35% sucrose and flash frozen in liquid nitrogen. X-ray

diffraction data were collected at beamline PROXIMA 2 A (SOLEIL synchrotron) and processed and scaled with the aP_scale module in autoPROC (Global Phasing Limited)[35], using STARANISO to apply anisotropic correction to the data[36].

*MglA mixed form*: A unique crystal was obtained with MglA-GTPγS using a sparse matrix screen (Classics suite, Qiagen) in 0.1 M HEPES pH 7.5 and 70% MPD, by hanging drops dispensed with a Cartesian crystallization robot. The crystal was cryoprotected with 10% glycerol and flash frozen before data collection at beamline PROXIMA 1 (SOLEIL synchrotron). Diffraction data were processed using XDS[34] and scaled with Aimless[37].

*Unbound MglB*: Initial crystallization conditions were identified using sparse matrix screens with a Cartesian robot and optimized to produce diffracting crystals. Two crystal forms were obtained, one in 20% isopropanol and 10% PEG 4000 which was used to derive a low-resolution envelope (form 1), and one in 0.1 M MES pH 6, 0.15 M ammonium sulfate and 15% (w/w) PEG 4000 which was used for structure determination (form 2). Crystals were cryoprotected with 10% glycerol and flash frozen before data collection at beamline PROXIMA 1 at SOLEIL synchrotron. Diffraction data were processed and scaled using the XDS package[34].

*MglA-GTPγS/MglB complex*: The complex was prepared by incubating equimolar amounts of MglA and MglB with a 20× molar excess of GTPγS for 30 min at room temperature followed by purification by size exclusion chromatography using a Superdex 75 column and concentration at 20 mg/mL with a 5× molar excess of GTPγS. Initial crystallization conditions were obtained with the Protein Complex Suite (Qiagen) screened by sitting drop-vapor diffusion at 18 ° C using a Mosquito robot and were optimized in hanging drops by mixing 1 μL of the sample with 1 μL of the precipitant. Diffracting crystals grew in 0.2 M HEPES pH 7 and 16% PEG 4000. Crystals were cryoprotected by transfer to a solution containing 0.2 M HEPES pH 7 and 35% PEG 4000 and flash frozen in liquid nitrogen. X-ray diffraction data were collected at beamline ID30-B at the European Synchrotron Radiation Facility (Grenoble, France). Diffraction data were processed and scaled with autoPROC (Global Phasing Limited)[35], using STARANISO to apply anisotropic correction to the data[36].

**Structure determination and refinement**. Structures of MglA-GDP, MglA-GTPγS and the MglA mixed form were solved by molecular replacement using Phaser[38] from the PHENIX suite[39]. The structure of *T. thermophilus* MglA-GDP (PDB 3T1O [12]) was used as a search model to solve the MglA-GDP structure. The refined structure of *M. xanthus* MglA-GDP was then used as a search model for subsequent molecular replacements. The structure of the MglA-GTPγS-MglB complex was solved by molecular replacement using the refined structures of MglB and MglA-GTPγS as search models.

No molecular replacement solution could be obtained for unbound MglB form 1 or form 2 crystals using *T. thermophilus* MglB as a search model. Alternatively, we used both crystals forms for molecular replacement with AMoRe[40] followed by density modification[41,42]. First, molecular replacement was carried out with monoclinic (P2$_1$) data (form 1, 15-4 Å) by using a monomeric multiple model (nine superimposed structures of *T thermophilus* MglB monomers from PDB entries 1J3W, 3T1R, 3T1S with pruned side-chains) and by applying rotational noncrystallographic symmetry (NCS) constraints based on self-rotation function peaks. The four monomeric models placed in the P2$_1$ asymmetric unit appeared in an arrangement similar to the tetrameric structure of *T. thermophilus* MglB, though with imperfect 222 symmetry. Successive density modification including fourfold NCS averaging and solvent flattening and phase extension to 3.3 Å provided us with electron density maps of mediocre quality, perhaps due to incompleteness of data though this was not further investigated, but sufficient to distinguish and carve out the average density of an MglB monomer (Supplementary Fig. 2a). This density was then used as a molecular replacement probe to phase the P2$_1$2$_1$2$_1$ orthorhombic data (form 2 crystal). Twenty copies of the monomer, corresponding to five tetramers, were placed in the P2$_1$2$_1$2$_1$ asymmetric unit using data in the 15 −4 Å range and by applying NCS constraints based on self-rotation function peaks. Successive density modifications by 20-fold NCS averaging and solvent flattening and phase extension to 2.4 Å yielded good quality and interpretable density maps for all chains (Supplementary Fig. 2b).

Refinements were performed by multiple rounds of refinements with PHENIX[43] and manual fitting in Coot[44]. Crystallographic statistics are given in Supplementary Table 1. Structural data have been deposited with the Protein Data Bank with accession codes given in Table 1.

**Nucleotide exchange and GAP kinetics assays**. Nucleotide exchange kinetics were measured by monitoring the decrease in fluorescence following mant-GDP dissociation and replacement by GTP added in fivefold molar excess ($\lambda_{Ex} = 360$ nm, $\lambda_{Em} = 440$ nm).

GAP kinetics were measured by fluorescence, using an engineered bacterial phosphate-binding protein (PBP) (kind gift of Martin Webb, UK) to detect inorganic phosphate produced by GTP hydrolysis ($\lambda_{Ex} = 430$ nm, $\lambda_{Em} = 465$ nm)[25]. After expression and purification, PBP was labeled with N-[2-(1-maleimidyl)ethyl]-7-(diethylamino)coumarin-3-carboxamide (MDCC). GAP experiments were performed at 25 °C using 5 μM (Fig. 3a, b) or 10 μM (Fig. 3c, d) of the PBP sensor, 1 μM of MglA and a range of MglB concentrations as indicated. Fluorescence measurements were done in a FlexStation Multi-Mode Microplate reader. $k_{obs}$ values were determined from a mono-exponential fit. $k_{cat}/k_m$ were determined by

linear regression from $k_{obs}$ values measured over a range of GAP concentrations following the Michaelis–Menten model.

All experiments were done in triplicate.

**SEC-MALS.** For size exclusion chromatography coupled to multiangle light scattering (SEC-MALS) analyses, MglB was prepared at 3 mg/mL in a buffer containing 20 mM Tris pH 7.5, 50 mM NaCl, 10 mM MgCl₂. The same buffer was used as the mobile phase for SEC using a Superdex 75 10/300 GL column on a Shimadzu HPLC. Multiangle light scattering was detected with a MiniDAWN TREOS light scattering module and a refractometer Optilab T-rEX (Wyatt Technology).

**Circular dichroism.** Measurements were performed at 25 °C using a Jobin-Yvon Marker IV high sensitivity dichrograph. The MglA sample was dialyzed in 20 mM phosphate buffer at pH 8 and kept at 4 °C. Measurements were recorded at 0, 10 and 30 min after placing the sample at 25 °C. Far-UV spectra were collected from 190 to 260 nm with 1 nm steps in a 0.1 cm path-length quartz cell. Three scans were averaged and corrected by subtracting a buffer spectrum.

**Nucleotide content determination.** MglA-GTP was incubated for 1 h on ice or at 25 °C, denatured by addition of methanol, kept at −20 °C for 2 h and then centrifuged to remove the precipitated protein. The nucleotides in the soluble fractions were analyzed by ion exchange in a MonoQ column recording absorbance at 254 nm. Pure GTP and GDP diluted in the same buffer were analyzed as a reference.

**SAXS data analysis and structural modeling.** MglB SAXS data were collected using the inline HPLC-coupled SAXS instrument at SWING beamline (SOLEIL Synchrotron, France). Six hundred micrograms MglB in a 40 µL volume (15 mg/mL) was injected into a size exclusion chromatography column (SEC-3 300 Å, Agilent Technologies, Inc.) equilibrated with elution buffer (20 mM Tris pH 8.0, 150 mM NaCl and 1 mM DTT), prior to SAXS data acquisition. Data reduction to absolute units, frame averaging, and subtraction were done using the FOXTROT program (synchrotron SOLEIL). Frames corresponding to the high-intensity fractions of the peak and having constant radius of gyration ($R_g$) were averaged. All SAXS data analyses were performed with programs from the ATSAS package[45]. $R_g$ was evaluated using the data within the range of Guinier approximation $sR_g < 1.3$ and by the Guinier Wizard and Distance Distribution Wizard. The maximum distance $D_{max}$ was estimated with PRIMUS and refined by trial and error with GNOM. The distance distribution functions $P_{(r)}$ were calculated with GNOM. The dimensionless Kratky plot was calculated by plotting $(qR_g)^2I_{(q)}/I_{(0)}$ against $qR_g$. The molecular weight was estimated by PRIMUS Molecular Weight wizard. The fit between scattering experimental amplitudes and amplitudes calculated from the crystal structure of the MglB dimer was calculated with CRYSOL. Ten independent ab initio models were calculated with GASBOR, using data at $q = 0.25$ and imposing P2 symmetry, which were compared with SUPCOMB and clustered with DAMCLUST. The consensus model was represented by the lowest Normalized Spatial Discrepancy (NSD), which was determined with DAMSEL. The flexible C-terminal fragments in each MglB monomer were modeled with MultiFoXS[46]. The resulting models were clustered into four ensembles, yielding an excellent fit to the experimental SAXS data. The SAXS data have been deposited with the SAXSDB database under the accession code SASDET9. SAXS statistics are given in Supplementary Table 2.

**Liposome cosedimentation.** All lipids are natural lipids from Avanti Polar Lipids. Liposomes were prepared with 76% phosphatidylethanolamine (PE), 4.9% phosphatidylglycerol (PG), 9.3% cardiolipin (CL), 6.5% phosphatidylserine (PS), and 3% lysophosphatidylcholine (LPC) in 50 mM Tris buffer pH 7.5 with 220 mM sucrose. After five cycles of freezing and thawing, liposomes were extruded through a 0.2 µm polycarbonate filter. Sucrose was removed by dilution in a buffer containing 50 mM Tris pH 7.5 and 120 mM NaCl, followed by centrifugation at $390,880 \times g$ and resuspension in the same buffer. Cosedimentation assays were performed by incubating proteins (1 µM) with liposomes (1 mM) at room temperature for 10 min, followed by centrifugation at $35,5040 \times g$ for 20 min. Controls were prepared without liposomes. Supernatants were recovered and pellets were resuspended in the original buffer volume. All samples were analyzed by SDS-PAGE. All experiments were done in triplicate.

**Bacterial strains and genetic constructs.** Strains, plasmids and primers used for this study are listed in Supplementary Tables 3–5. In general, *M. xanthus* strains were grown at 32 °C in CYE-rich media[47]. Plasmids were introduced in *M. xanthus* by electroporation. Complementation, expression of the fusion and mutant proteins was obtained by ectopic integration of the genes of interest at the Mx8-phage attachment site[8] under the control of their own promoter in appropriate deletion backgrounds. *E. coli* cells were grown under standard laboratory conditions in Luria−Bertani broth supplemented with antibiotics, if necessary.

**Expression of MglB and MglB³ᴹ.** MglB and MglB³ᴹ were expressed in the mglB deletion strain. The pSWU19 *MglB³ᴹ* for complementation of the *mglB* deletion

was constructed by amplifying the *mglB* coding sequence. The mutated fragment was cloned into the pSWU19 vector by the one-step sequence- and ligation-independent cloning (SLIC) method[48].

**Construction of MglB- and MglB³ᴹ-neon green fusions.** MglB-NG and derivatives were expressed by complementing the *mglB* deletion mutant[8] by integration of pSWU19 *mglB-ng/mglB³ᴹ-ng* at the Mx8 Phage attachment site. For this, the MglB/MglB³ᴹ (using pET28 *MglB³ᴹ* as a template, Table S4) and mNeon-green encoding sequences were amplified by PCR and mixed for cloning into the pSWU19 vector by the SLIC method.

**Motility assays on plate.** For soft-agar motility assays[47] the cells were grown up to an OD between 0.4 to 0.8 and concentrated at OD = 5 then spotted (10 µL) on CYE 0.5% (soft). Colonies were photographed after 48 h.

**Fluorescence imaging and fluorescence intensity measurements.** For phase-contrast and fluorescence microscopy, cells from exponentially growing cultures were concentrated to an OD = 2 by centrifugation of 1 mL of culture and resuspended in the corresponding volume of TPM buffer (10 mM Tris-HCl, pH 7.6, 8 mM MgSO₄, and 1 mM KH₂PO₄). Then, a drop of 2 µL was deposited on a coverslip and covered with a 1.5 % agar pad with TPM buffer. Microscopic analysis was performed using an automated and inverted epifluorescence microscope TE2000-E-PFS (Nikon, France) with a 100×/1.4 DLL objective and a CoolSNAP HQ2 camera (Photometrics). All fluorescence images were acquired with a minimal exposure time to minimize bleaching and phototoxicity effects.

**Cell tracking.** Image analysis was performed with MicrobeJ using a FIJI-based tracking procedure developed for bacteria[49]. Cells were detected with MicrobeJ by thresholding the phase-contrast images after stabilization. Cells were tracked using MicrobeJ on a minimum of 40 frames by calculating all object distances between two consecutive frames and selecting the nearest objects. The computed trajectories were systematically verified manually and, when errors were encountered, the trajectories were removed. Analysis of the trajectories, distance to origin and MSD calculations were performed with MicrobeJ. For each strain, at least two biological replicates acquired independently were analyzed.

**Cluster counting.** Image analysis was performed with FIJI with the cell counter plugin and under MicrobeJ with the Maxima detection system. For cluster detection, all fluorescence images were acquired with a 1 s exposure time for optimal signal-noise ratio.

**Western blots.** Samples were grown at 32 °C in CYE medium to an optical density (OD) at 600 nm (OD600) of 0.4−1, a volume of culture equivalent to 1 mL was centrifuged for 5 min at $4500 \times g$. The pellet corresponding to whole cells was resuspended in SDS-PAGE loading buffer containing β-mercaptoethanol to 10 OD600 units and heated for 10 min at 99 °C. Proteins samples equivalent 1 OD600 unit were separated by SDS-PAGE. Electrophoresis was performed at 180 V for 50 min at room temperature using 10% SDS-polyacrylamide gel. For western blotting, proteins were transferred from gels onto nitrocellulose membranes. The membranes were blocked during 1 h at room temperature in Tris-buffered saline (pH 7.6), 5% milk, 0.2% Tween 20 (for MglA) or in Tris-buffered saline (pH 7.6), 2% milk, 0.2% Tween 20 (for MglB) and incubated with primary antibodies directed against MglA (dilution at 1:5000) or MglB (dilution at 1:2500) in blocking buffer overnight at 4 °C. After two washings of 5 min with Tris-buffered saline (pH 7.6), 0.2% Tween 20 membrane were incubated with Goat Anti-Rabbit IgG (H + L)-HRP Conjugate (#1706515, Biorad, dilution at 1:10000) in the respective blocking buffers. The peroxidase reaction was developed by chemiluminescence (SuperSignal™ West Pico Chemiluminescent Substrate #34080 Thermo Scientific™) scanned and analyzed with ImageQuant LAS 4000 and TL analysis software (GE Healthcare Life Sciences).

**Reporting summary.** Further information on research design is available in the Nature Research Reporting Summary linked to this article.

## Data availability

Data supporting the findings of this manuscript are available from the corresponding author upon reasonable request. A reporting summary for this Article is available as a Supplementary Information file. The source data underlying Fig. 4e and Supplementary Figs. 3b, 4b, e are provided as a Source Data file. Coordinates and structure factors of X-ray crystallography structures have been deposited in the Protein Data Bank under accession codes 6HJO, 6H17, 6H35, 6H5B and 6HJM. SAXS data have been deposited in the SASBDB database under accession code SASDET9.

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

## Acknowledgements

This work was supported by grants from the ANR to J.C. and T.M. (ANR-15-CE13-0006) and to J.C. (ANR-14-CE09-0028). We are grateful to Armelle Vigouroux (LEBS, CNRS, Gif-sur-Yvette) for preliminary crystallization experiments, and Michel Franco (IPMC, CNRS, Sophia-Antipolis) for suggesting the liposome composition. We thank the scientific staff at X-ray crystallography beamlines PROXIMA1 and PROXIMA 2 and SAXS beamline SWING (SOLEIL synchrotron, Gif-sur-Yvette, France) and X-ray crystallography beamline ID-29 (ESRF synchrotron, Grenoble, France) for making the beamlines available to us and for expert advice.

## Author contributions

C.G. designed, performed and analyzed the biochemistry experiments and performed the crystallographic analysis of MglA-GTPγS and the MglA-MgB complex. S.L. and J. H. performed and analyzed the Myxococcus motility experiments. P.F.V. performed the crystallographic analysis of MglA-GDP, the mixed MglA form and unbound MglB. W.Z. performed the SAXS analysis. J.N. and S.T. solved the structure of unbound MglB by molecular replacement. T.M. and J.C. conceived the project and

supervised the study. C.G., T.M. and J.C. wrote the manuscript with input from all authors.

## Competing interests

The authors declare no competing interests.
