## [Peer Review File · Nature Communications]

Reviewers' Comments:

Reviewer #1:

Remarks to the Author:

This work from the Cherfils lab explores a 3-state GTPase switch, regulating pole-to-pole oscillation of the motility machinery in *M. xanthus*. The authors use structural biology, in vitro fluorescence kinetics and in vivo motility assays to characterise a positive feedback mechanism required for the reversal of the lagging to the leading pole. The centre of this study is a novel hybrid state of GTP bound MglA, which combines features of its active and inactive conformation. In this state the GTPase is insensitive to its GAP MglB. An interesting observation is that this hybrid GTPase state transforms into GAP sensitive MglA upon interaction with the GDP bound form of MglB, thus defining a positive feedback loop. Overall this work is technically sound. Valid conclusions are drawn from a coherent repertoire of results. In particular the structural work is comprehensive which includes 5 crystal structures and some SAXS data. The MglB structure is interesting from a crystallographic point of view which was solved by MR using data sets from crystals comprising 20 molecules in the AU and thus setting a benchmark for molecular replacement using crystals with large numbers of copies. The authors deliver an exciting study which shed light on the regulation of motility in gliding bacteria and expands the conceptual mode of action of small GTPases in general. I therefore support publication in Nature Communications.

Main points:

- 1) An obvious weak point of this manuscript is the lack of information about the potential mechanism of interconversion of the two MglA-GTP/MglA-GTP* species. The authors describe that spontaneous conversion of MglA-GTP into the hybrid form occurs within 1 hour but pole to pole oscillating dynamics of MglA requires just a couple of minutes. Does this suggest the existence of additional factors which catalyse conversion? I would not expect further experimental evidence which elucidates the mechanism of this interconversion but a more detailed discussion how the observed kinetic fits to the in vivo oscillating dynamics would be an asset.
- 2) Since most claims of this manuscript depend on finding an unusual structure of MglA-GDP Pi/MglA-GDP, the authors should reassure the reader that there is no ambiguity in data interpretation by providing supplemental figures of omit maps showing close-up views of GDP-Pi and the switch regions.
- 3) To visualize the relative arrangement of the 20 copies of the MglB structure, a supl. figure which shows the backbone of all molecules in the AU would be helpful.

Minor points:

- 1) Figure S3D: According to the figure legend there is a scale bar which I can't find in the figure.

Reviewer #2:

Remarks to the Author:

The manuscript by Galicia et al. reports incisive new findings and insights into the oscillatory nature of small GTPase proteins and their complexes to elicit directed movement in bacteria. The study is elegant, interdisciplinary, and provides a structural and mechanistic framework to help move the field forward.

Specific comments:

- (1) Despite the plethora and importance of the findings the abstract does not fully capture the essence of the findings and their positioning in biology. For instance, a general description of the

importance of directed movement in bacteria is conspicuously absent. The authors are advised to drastically revise their abstract.

(2) For readers who are not familiar with the activation principles of small GTPases, it would be useful to include a schematic representation of what one should expect from active, autoinhibited, and inactive conformation of loops or switch regions. Figure 1 could benefit from such a schematic representation. In this way, the reader will also better appreciate the unique capture of the mixed state.

(3) The maintenance of structural disorder in the C-terminal segment of bound and unbound MglB is inadequately rationalized. The authors state that this region is conserved, yet they do not provide any functional implications. A recent study by Keul et al. *Nature*. 2018 Nov;563(7732):584-588. doi: 10.1038/s41586-018-0699-5 might prove relevant/useful.

(4) Figure 5 does not quite provide the essence of the mechanistic aspects of the work and its implications to a reader who will be looking at the figure without having read the manuscript. The authors might want to strike a better balance between what is provided in the figure legend and the illustration.

(5) It would be useful to include a supplementary figure to illustrate and summarize the creative and elaborate structure solution scheme that led to the crystal structure of the MglA-GTPgS-MglB complex. The figure should include examples of electron density from key steps, in particular the steps after application of NCS-averaging procedures.

(6) Table S1 could be fused with Table 1. Also, please change the representation of R-work/R-free from percentage to decimal. e.g. 21.4/23.8 to 0.214/0.238 for the MglA-GDP complex

Responses to Reviewers' comments.

We are grateful to our reviewers for their supportive and constructive comments, which have been most valuable to help us improve our manuscript. We have addressed their queries very carefully and we believe that we have been able to provide appropriate answers to all of them. All changes are highlighted in red in the manuscript, and described in detail below.

Reviewer #1 (Remarks to the Author):

*This work from the Cherfils lab explores a 3-state GTPase switch, regulating pole-to-pole oscillation of the motility machinery in *M. xanthus*. The authors use structural biology, in vitro fluorescence kinetics and in vivo motility assays to characterise a positive feedback mechanism required for the reversal of the lagging to the leading pole. The centre of this study is a novel hybrid state of GTP bound MglA, which combines features of its active and inactive conformation. In this state the GTPase is insensitive to its GAP MglB. An interesting observation is that this hybrid GTPase state transforms into GAP sensitive MglA upon interaction with the GDP bound form of MglB, thus defining a positive feedback loop. Overall this work is technically sound. Valid conclusions are drawn from a coherent repertoire of results. In particular the structural work is comprehensive which includes 5 crystal structures and some SAXS data. The MglB structure is interesting from a crystallographic point of view which was solved by MR using data sets from crystals comprising 20 molecules in the AU and thus setting a benchmark for molecular replacement using crystals with large numbers of copies. The authors deliver an exciting study which shed light on the regulation of motility in gliding bacteria and expands the conceptual mode of action of small GTPases in general. I therefore support publication in Nature Communications.*

Main points:

1) An obvious weak point of this manuscript is the lack of information about the potential mechanism of interconversion of the two MglA-GTP/MglA-GTP species. The authors describe that spontaneous conversion of MglA-GTP into the hybrid form occurs within 1 hour but pole to pole oscillating dynamics of MglA requires just a couple of minutes. Does this suggest the existence of additional factors which catalyse conversion? I would not expect further experimental evidence which elucidates the mechanism of this interconversion but a more detailed discussion how the observed kinetic fits to the in vivo oscillating dynamics would be an asset.*

The reviewer makes a very valid point. *In vivo*, we certainly envision that cellular factors accelerate the MglA-GTP to MglA-GTP* transition, a minima by binding preferentially to MglA-GTP* at the pole which will shift the MglA-GTP <-> MglA-GTP* equilibrium towards this state, as was in fact already mentioned in the discussion. Importantly, even if the conversion of MglA-GTP to MglA-GTP* is relatively slow, the resensitization of MglA-GTP* by the feedback effect occurs within seconds (see Fig 3C), which is fast enough to make polar MglA available for its relocalization to the opposite pole and matches the relocalization kinetics observed *in vivo*. We have reworded this section to make this point clearer (lines 359-364)

2) *Since most claims of this manuscript depend on finding an unusual structure of MglA-GDP Pi/MglA-GDP, the authors should reassure the reader that there is no ambiguity in data interpretation by providing supplemental figures of omit maps showing close-up views of GDP-Pi and the switch regions.*

We have added a new supplementary Figure (Figure S1 A-C) showing omit maps of the switch 1, switch 2 and nucleotide in the mixed MglA structure, which shows that all electron density is very well defined.

Please note that, accordingly, previous supplementary Figures S1 to S3 are now labeled supplementary Figures S2 to S4

3) *To visualize the relative arrangement of the 20 copies of the MglB structure, a supl. figure which shows the backbone of all molecules in the AU would be helpful.*

We have added a supplementary panel S2C in supplementary Figure S2 (previously S1) that shows the 20 MglB monomers in the asymmetric unit and highlights their arrangement as 5 tetramers.

Minor points:

1) *Figure S3D: According to the figure legend there is a scale bar which I can't find in the figure.*

The scale bar has been added.

Reviewer #3 (Remarks to the Author):

The manuscript by Galicia et al. reports incisive new findings and insights into the oscillatory nature of small GTPase proteins and their complexes to elicit directed movement in bacteria. The study is elegant, interdisciplinary, and provides a structural and mechanistic framework to help move the field forward.

Specific comments:

(1) *Despite the plethora and importance of the findings the abstract does not fully capture the essence of the findings and their positioning in biology. For instance, a general description of the importance of directed movement in bacteria is conspicuously absent. The authors are advised to drastically revise their abstract.*

We have revised the abstract, which hopefully now better highlights our findings. Please note that there is a limit count of 150 words, which is why some aspects of the biology could not be mentioned. Instead, we have added a sentence and a new reference to a recent review in the introduction (Reference 5), which points to the importance of directed movement for the multicellular behaviour of the bacteria (Lines 54-55)

(2) *For readers who are not familiar with the activation principles of small GTPases, it would be useful to include a schematic representation of what one should expect from active, autoinhibited, and inactive conformation of loops or switch regions. Figure 1 could benefit from such a schematic representation. In this way, the reader will also better appreciate the unique capture of the mixed state.*

We appreciate this suggestion and have added a new panel in Figure 1 (panel 1A) showing a schematic representation of the three MglA conformations. Please note that accordingly, the panels in Figure 1 previously labeled (A-I) are now labeled (B-J).

(3) The maintenance of structural disorder in the C-terminal segment of bound and unbound MglB is inadequately rationalized. The authors state that this region is conserved, yet they do not provide any functional implications. A recent study by Keul et al. Nature. 2018 Nov;563(7732):584-588. doi: 10.1038/s41586-018-0699-5 might prove relevant/useful.

We thank this reviewer for alerting us on this important study. Indeed, the entropic determinants of intrinsically disordered regions are increasingly being shown to support a remarkable variety of functional properties such as substrate recognition, enzyme efficiencies, membrane curvature sensing and clustering on membranes or through liquid-liquid phase separations. It is thus plausible that one or several such properties are supported by the disordered C-termini of MglB and contribute to its clustering and function at the lagging pole, which remain to be investigated.

We have modified this section accordingly, including the addition of 4 additional references each highlighting such properties, including the one pointed out to us by this reviewer (references 29-30-31-34, lines 389-394)

(4) Figure 5 does not quite provide the essence of the mechanistic aspects of the work and its implications to a reader who will be looking at the figure without having read the manuscript. The authors might want to strike a better balance between what is provided in the figure legend and the illustration.

We thank this reviewer for this suggestion. We have added a graphical legend to Figure 5 which now explains visually what T, B, T* and D mean. We believe that this schema should be now mostly auto-explanatory without having to refer to the legend or the manuscript, where the details can be found.

(5) It would be useful to include a supplementary figure to illustrate and summarize the creative and elaborate structure solution scheme that led to the crystal structure of the MglA-GTPγS-MglB complex. The figure should include examples of electron density from key steps, in particular the steps after application of NCS-averaging procedures.

We appreciate the interest of this reviewer in this original molecular replacement procedure. We have added two new panels (panels A-B) to supplementary Figure 2 (previously Figure S1) which show the NCS-averaged maps at each step, highlighting the map improvement.

Please note that previous panels A-I in this supplementary Figure S2 are now panels D-L due to the addition of the NCS-averaged maps (panels A-B) and a view of the MglB asymmetric unit (panel C; please see above our answers to reviewer 1).

(6) Table S1 could be fused with Table 1. Also, please change the representation of R-work/R-free from percentage to decimal. e.g. 21.4/23.8 to 0.214/0.238 for the MglA-GDP complex.

The change to decimal has been made. We would however prefer to keep the summary Table 1, as it provides a brief summary of the crystallographic data in the main part of the manuscript.

Reviewers' Comments:

Reviewer #1:

Remarks to the Author:

The authors have satisfactorily addressed my comments. Implementation of the requested figures and a more detailed discussion about the kinetics under in vivo and in vitro conditions have improved the manuscript which can now be recommended for publication.

Reviewer #2:

Remarks to the Author:

Dear Editor,

the revised manuscript addresses my remarks well and is of high quality.